# Landscape genomic prediction for restoration of a *Eucalyptus* foundation species under climate change

Megan Ann Supple[1,2]*, Jason G Bragg[1,3], Linda M Broadhurst[4], Adrienne B Nicotra[1], Margaret Byrne[5], Rose L Andrew[6], Abigail Widdup[1], Nicola C Aitken[1], Justin O Borevitz[1,7]

[1]Research School of Biology, The Australian National University, Canberra, Australia; [2]Department of Ecology and Evolutionary Biology, University of California, Santa Cruz, Santa Cruz, United States; [3]National Herbarium of New South Wales, The Royal Botanic Gardens and Domain Trust, Sydney, Australia; [4]Centre for Australian National Biodiversity Research, Commonwealth Scientific and Industrial Research Organisation (CSIRO), National Research Collections and Facilities, Canberra, Australia; [5]Biodiverstiy and Conservation Science, Department of Biodiversity, Conservation and Attractions Western Australia, Bentley, Australia; [6]School of Environmental and Rural Science, University of New England, Armidale, Australia; [7]Centre of Excellence in Plant Energy Biology, The Australian National University, Canberra, Australia

**Abstract** As species face rapid environmental change, we can build resilient populations through restoration projects that incorporate predicted future climates into seed sourcing decisions. *Eucalyptus melliodora* is a foundation species of a critically endangered community in Australia that is a target for restoration. We examined genomic and phenotypic variation to make empirical based recommendations for seed sourcing. We examined isolation by distance and isolation by environment, determining high levels of gene flow extending for 500 km and correlations with climate and soil variables. Growth experiments revealed extensive phenotypic variation both within and among sampling sites, but no site-specific differentiation in phenotypic plasticity. Model predictions suggest that seed can be sourced broadly across the landscape, providing ample diversity for adaptation to environmental change. Application of our landscape genomic model to *E. melliodora* restoration projects can identify genomic variation suitable for predicted future climates, thereby increasing the long term probability of successful restoration.
DOI: https://doi.org/10.7554/eLife.31835.001

**\*For correspondence:**
megan.a.supple@gmail.com

**Competing interests:** The authors declare that no competing interests exist.

## Introduction

Species around the globe face rapidly changing environments, often in combination with habitat loss and fragmentation. These factors are expected to have a negative impact on biodiversity (*Lindenmayer et al., 2010*). Three processes enable species to survive altered conditions: migration, adaptation, and phenotypic plasticity (*Aitken and Whitlock, 2013*; *Aitken et al., 2008*; *Hoffmann et al., 2015*; *Nicotra et al., 2010*). An important conservation strategy is to assist these natural processes to help build more resilient communities. We can help populations to become better adapted to future environmental conditions by assisting migration of gene pools across the landscape (*Aitken and Whitlock, 2013*; *Aitken et al., 2008*). We can aid populations to survive in situ by ensuring that sufficient genomic variation exists for adaptation to changing environments

**eLife digest** Yellow box, or *Eucalyptus melliodora*, is an emblematic Australian tree that is essential to many native ecosystems. Some of these environments are now critically endangered, and replanting yellow box trees is one of the first steps to try to restore them.

However, it can be difficult for reforestation practitioners to decide where to collect the seeds they will use to replant an area. They have to select seeds that carry the genetic information that gives the trees the best chances of surviving now and in the future. This is a complex task because climate change creates fast-changing environments.

Here, Supple et al. collect genetic material from 275 *E. melliodora* trees at 36 different sites. Genetic analyses show that the yellow box trees from these sites exchange their genetic material and do not form isolated populations. This means that the seeds do not need to be sourced from near the reforestation site, but can be collected from areas much further away. This results in higher quality seeds for reforestation because seeds from more sites will include more genetic diversity.

Supple et al. then use information about the local climate, such as temperature and rain levels, at the sites where the samples were gathered to create a model that describes the relationship between genetic, geographical, and environmental factors. This helps identify which sites produce the seeds that will grow best under particular conditions. In addition, the seedlings from these sites are grown in the laboratory to see how well they fare in different types of environments. It therefore becomes possible to match a reforestation site with the seeds that will thrive in the future climate of the area.

Sharing this knowledge with conservationists will help to guide their replanting strategies for *E. melliodora*. The method can also be applied to other plant species and restoration projects, so it could ultimately shape resilient ecosystems that can cope with climate change.

DOI: https://doi.org/10.7554/eLife.31835.002

(*Hoffmann et al., 2015*). We can enable individuals to respond to a greater range of environments by conserving existing phenotypic plasticity (*Nicotra et al., 2010*).

Seed sourcing during landscape restoration provides an ideal opportunity to apply scientific knowledge to enable these key processes and improve conservation outcomes (*Broadhurst et al., 2008*; *Prober et al., 2015*). For example, seed sources can be selected to restore historical patterns of gene flow across fragmented landscapes and to incorporate high levels of available genomic diversity. If plasticity varies among populations, seed can be selected to augment the phenotypic plasticity of individuals at restoration sites. Seed sources can also be matched with current or predicted future climates, enabling assisted migration to favorable environments (*Aitken and Whitlock, 2013*; *Williams et al., 2014*).

Historically, restoration has often focused on geographically restricted local sources of seed under the premise that this would improve restoration outcomes by reducing the risk of maladaptation to local conditions and by preventing outbreeding depression (*Broadhurst et al., 2008*). However, there are several potential drawbacks to this narrow local focus. In a fragmented system, narrow local seed sourcing reduces the number of potential source populations, thereby reducing the pool of available genetic material. This reduced gene pool may result in inbreeding depression in future generations, especially if combined with small population size (*Broadhurst et al., 2008*). Obtaining seed from a wider geographical area can increase genomic and phenotypic diversity, thereby increasing the ability of the species to survive in situ (*Broadhurst et al., 2008*). Additionally, the focus on maintaining local adaptation assumes a static environment, not the rapidly changing environments that occur today. When local conditions change, traits and genes that have conferred an advantage in the past may not be suitable in future environments. In recent years, climate adjusted provenancing has been proposed, providing a seed sourcing strategy that focuses on both genetic diversity and adaptability under predicted future conditions (*Byrne et al., 2013*; *Prober et al., 2015*). This strategic assisted migration of variation across the landscape can aid in the establishment of populations that are more adaptable to future environments (*Prober et al., 2015*).

To identify an appropriate seed sourcing strategy for a reforestation project, it is useful to characterize genomic variation in the target species with empirical data. These data can be used to infer

patterns of Isolation By Distance (IBD) and Isolation By Environment (IBE). IBD describes the correlation between genomic distance and geographic distance, which arises when gene flow occurs more often between populations that are in close geographic proximity. IBE describes the correlation between genomic distance and environmental distance, while controlling for geographic distance (*Wang and Bradburd, 2014*). IBE arises because environmental drivers can influence gene flow, so that migration rate is effectively modulated by the environment. This means IBE is detectable in genome-wide variation, and not just at loci mediating adaptation. Landscape genomic models can be generated that describe the relationship between genetic differentiation and both spatial and environmental distances (representing IBD and IBE). These predictive models can be used to optimize the genetic material selected for restoration and should improve long term outcomes (*Hoffmann et al., 2015*; *Williams et al., 2014*).

The extent of phenotypic plasticity in potential seed sources can be measured in growth assays of seedling traits across contrasting environmental conditions. The magnitude of the environmental response can be compared among maternal lines or populations and may identify populations that differ in their response to the environment. Such differing responses have been seen in some species of *Eucalyptus* (*Andrew et al., 2010*; *Byrne et al., 2013*; *McLean et al., 2014*), which typically have high levels of within-population genetic variation and moderate-high rates of outcrossing (*Byrne, 2008*).

*Eucalyptus melliodora* (A.Cunn. ex Schauer), commonly called yellow box, is an iconic Australian tree that is the subject of extensive restoration efforts across its distribution. It is a foundation species of a critically endangered ecological community: the White Box–Yellow Box–Blakely's Red Gum Grassy Woodland and Derived Native Grassland (*Department of Environment and Climate Change and Water, 2011*; *Department of the Environment and Heritage, 2006*; *Threatened Species Scientific Committee, 2006*). This woodland community exists in a fragmented landscape, with less than 5% of its original distribution remaining, mostly in small remnant patches (*Department of Environment and Climate Change and Water, 2011*; *Department of the Environment and Heritage, 2006*; *Threatened Species Scientific Committee, 2006*). Efforts to restore this endangered woodland community are ongoing and restoration practitioners are seeking scientific recommendations to improve seed sourcing. Climate change is an important consideration in seed sourcing decisions because species distribution modelling predicts that most eucalypts will need to shift their distributions considerably in response (*González-Orozco et al., 2016*). In particular, ecological niche modelling for *E. melliodora* predicts that by 2090 the species distribution will shift toward the southeast and suitable areas will decrease by 77% as a result of environmental changes (*Broadhurst et al., 2018*).

Here we survey genomic variation in 275 individuals from 36 sites across the present range of *E. melliodora*. To help determine an appropriate seed sourcing strategy, we fit the genotypic data to geographic distance and key environmental variables at the sites of origin. This enables characterization of isolation by distance across a broad area, providing an empirical estimate of 'local' for comparison with current practice for local provenancing. We also identify features of the abiotic environment that can further explain genomic differentiation after accounting for geographic distance. Additionally, we examine seedling growth under different simulated climate conditions to test for variation in growth traits and phenotypic plasticity both within and among sites. Our landscape genomic model, which can empirically define local provenances and identify variation suitable for predicted future climates, can help build resilient populations through scientifically based restoration.

## Results

### Genotyping by sequencing

We selected leaf material from 39 sites, sampling 3–10 trees per site (*Supplementary file 1*). For each sample we Illumina sequenced a Genotyping by Sequencing (GBS) library (*Elshire et al., 2011*) and used a reference alignment approach to call genotypes. We conducted a preliminary analysis based on 123,227 SNPs and removed 69 samples due to greater than 60% missing data. Visual examination of a cluster dendrogram of genomic distance between samples showed that technical replicates cluster closely together (*Figure 1—figure supplement 1*). A preliminary principal

coordinate analysis (PCoA) identified 19 samples that were strong genomic outliers (*Figure 1—figure supplement 2*), likely misidentified samples or recent hybrids. This result is consistent with minor morphological differences noted in these samples, as well as previous microsatellite work (*Broadhurst et al., 2018*). After removal of poor quality and geographic and genomic outlier samples, we re-ran the genotyping with the remaining 280 samples, resulting in 9,781 SNPs after filtering. A second preliminary PCoA identified an additional five outlier samples that we considered sufficiently differentiated from the main *E. melliodora* cluster to merit removal from downstream analyses (*Figure 1—figure supplement 3*). We removed these samples and reran the missing data filter. The final data set included 275 samples from 36 sites (*Figure 1A*), genotyped at 9,378 physically distinct SNPs (>300 bp apart).

## Genomic analyses

To help determine an appropriate seed sourcing strategy, we examined the effects that geography and environment have on the distribution of genomic variation across the landscape. The genomic

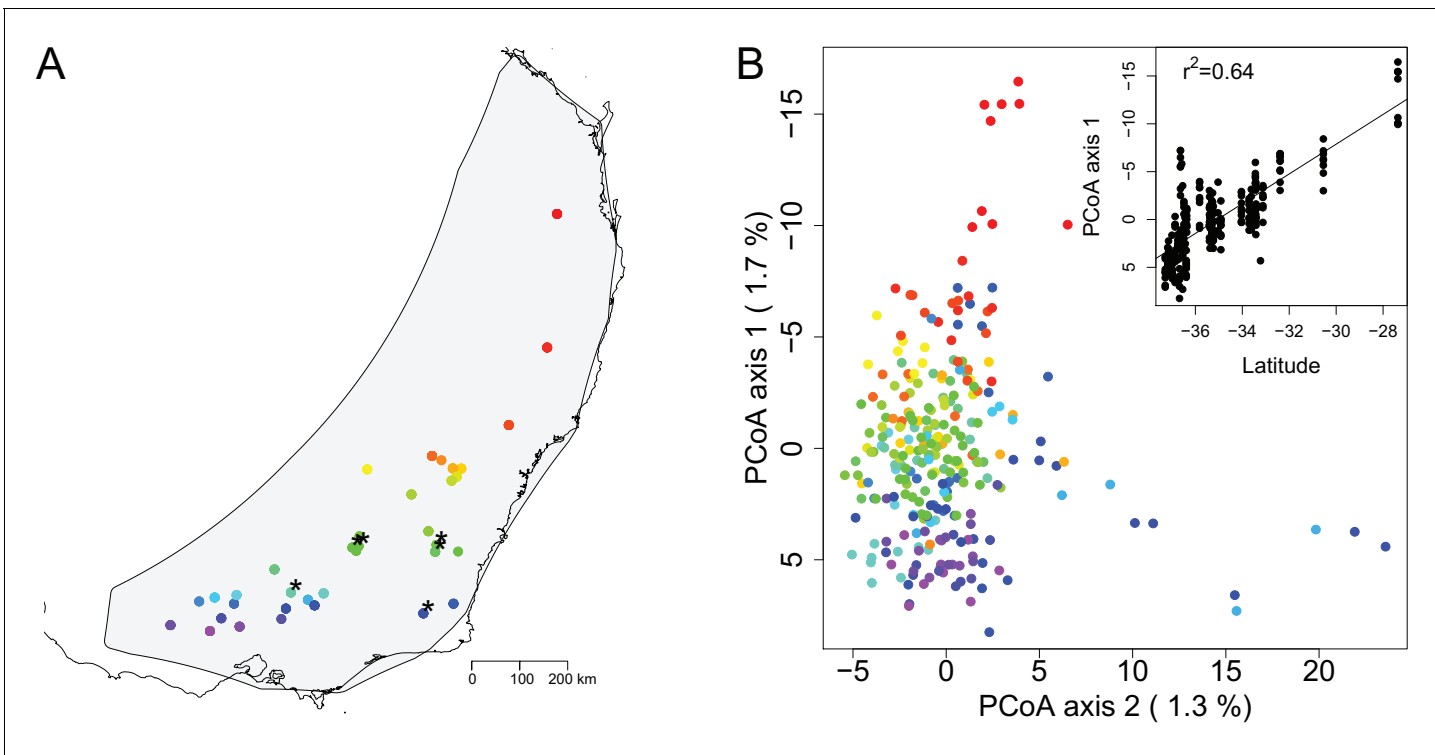

**Figure 1.** Map of sampling sites and PCoA of genomic distance between samples. (**A**) A map of the geographic locations of the 36 sampling sites in southeastern Australia. Sampling locations are indicated with dots color coded in a rainbow gradient based on latitude. Black asterisks indicate the six sites also used for growth chamber experiments. The gray background shading indicates the species distribution polygon. (**B**) Principal coordinate analysis of the genomic distance between individual samples. Samples are color coded by site to match the map. The percentage on each axis indicates how much of the genomic variation between individuals was explained by the axis. Note that PCoA axes 1 and 2 are switched from standard for easier visualization of the latitudinal gradient. The inset shows the regression of PCoA axis 1 against latitude.

DOI: https://doi.org/10.7554/eLife.31835.003

The following figure supplements are available for figure 1:

**Figure supplement 1.** Technical replicate dendrogram.
DOI: https://doi.org/10.7554/eLife.31835.004

**Figure supplement 2.** Species identification PCoA.
DOI: https://doi.org/10.7554/eLife.31835.005

**Figure supplement 3.** Outlier PCoA.
DOI: https://doi.org/10.7554/eLife.31835.006

**Figure supplement 4.** Site-level PCoA.
DOI: https://doi.org/10.7554/eLife.31835.007

analyses focused on the effects on the genome as a whole, rather than individual genes. The study of individual genes is beyond the scope of the current study.

The PCoA of genomic distance among samples showed continuous variation with little suggestion of discrete population structure (*Figure 1B*). This analysis, which was based on genomic data with no geographic information included, showed that the samples largely formed a single cluster, with the first PCoA axis correlating with latitude (*Figure 1B*). Outside of the main cluster, samples from the northernmost site separated out along the first PCoA axis (vertical axis) and a few samples from two other sites separated out along the second PCoA axis (horizontal axis). Together, the first two PCoA axes explained 3.0% of the genomic variation among individuals. The Mantel test, examining the correlation between geographic and genetic distance matrices, estimated that geographic distance between samples explained 2.3% of the variation in individual genomic distance, indicating weak, but statistically significant, isolation by distance (p=0.0001). We summarized genomic diversity between sampling sites using pairwise $F_{st}$. For all comparisons $F_{st}$ was low (mean $F_{st}$ = 0.04, sd = 0.02) (*Supplementary file 2*). The maximum $F_{st}$ of 0.10 occurs between sites 3 and 13, which are separated by over 1200 km. Similar to the individual-level PCoA of genomic distance among samples (*Figure 1B*), the site-level PCoA of $F_{st}$ between sampling sites also corresponded roughly to latitude (*Figure 1—figure supplement 4*). In contrast, the first two axes of the PCoA of $F_{st}$ between sampling sites explained a higher percentage of variation (37.1%). All sites with more than four individuals genotyped had similar levels of allelic diversity and expected heterozygosity (*Supplementary file 1*). Overall, these results highlight the low level of genetic structure over a large spatial scale in *E. melliodora*.

The site-by-site $F_{st}$ matrix was used to test for geographic and environmental correlations using generalized dissimilarity modelling (GDM) (*Ferrier et al., 2007*; *Fitzpatrick and Keller, 2015*; *Thomassen et al., 2011*). Of the 28 environmental variables considered for the model, we removed 12 variables because the single variable model explained less than 5% of the deviance (bioclimatic variables 2, 5, 6, 9, 10, 14, 17, 19; elevation; water at depth; Prescott Index; and MrVBF). We removed an additional nine variables due to high correlation and lower explanatory power than another remaining variable (bioclimatic variables 1, 4, 7, 12, 13, 15, 18; surface nitrogen; and surface phosphorus) (*Supplementary file 3*). We ran permutation testing on a model with the remaining seven variables, along with geographic distance. This highlighted an additional two variables with low statistical significance and low explanatory power. We removed these two variables (surface water and bioclimatic variable 8) from the final model. We also removed phosphorus at depth because, although it explained a substantial amount of genomic variation, the sampled sites were not well distributed across the range of phosphorus values.

As a result, we included four environmental variables in the final model: isothermality (bioclimatic variable 3), mean temperature of the coldest quarter (bioclimatic variable 11), precipitation of the wettest quarter (bioclimatic variable 16), and total soil nitrogen at 100–200 cm (nitrogen at depth) (*Figure 2*). The correlation coefficients between these variables were all less than 0.13, with the exception of the precipitation variable, which showed a moderate correlation with isothermality (r = 0.53) and nitrogen (r = 0.45) (*Supplementary file 3*). The GDM model with these four variables plus geographic distance explained 40% of the genetic differentiation ($F_{st}$) between sampling sites. The GDM model showed a positive non-linear relationship between environmental distance and genomic distance (*Figure 2A*). Visual examination of the genomic distances predicted from the model versus the observed values indicated the model had reasonable predictive power (*Figure 2B*). To quantify the predictive power of the GDM model, we used a cross validation approach by generating 1000 models with a random 30% of sampling sites removed. GDM proved satisfactory at predicting genomic differences between removed sites (cross validation correlation mean = 0.73, standard deviation = 0.12).

Geographic distance showed a non-linear relationship with genomic distance. The geographic spline predicted no genomic differentiation until close to 500 km, at which point an increase in geographic distance predicted an increase in genomic distance (*Figure 2C*). Randomly subsampling sites showed that the predicted genomic distance for large geographic distances was quite variable, but for sites less than 500 km apart, all iterations consistently predicted little genomic differentiation between sites (*Figure 2D*).

Of the four environmental variables, nitrogen at depth showed the strongest relationship with genomic distance, with changes in genomic distance predicted across the range of nitrogen values

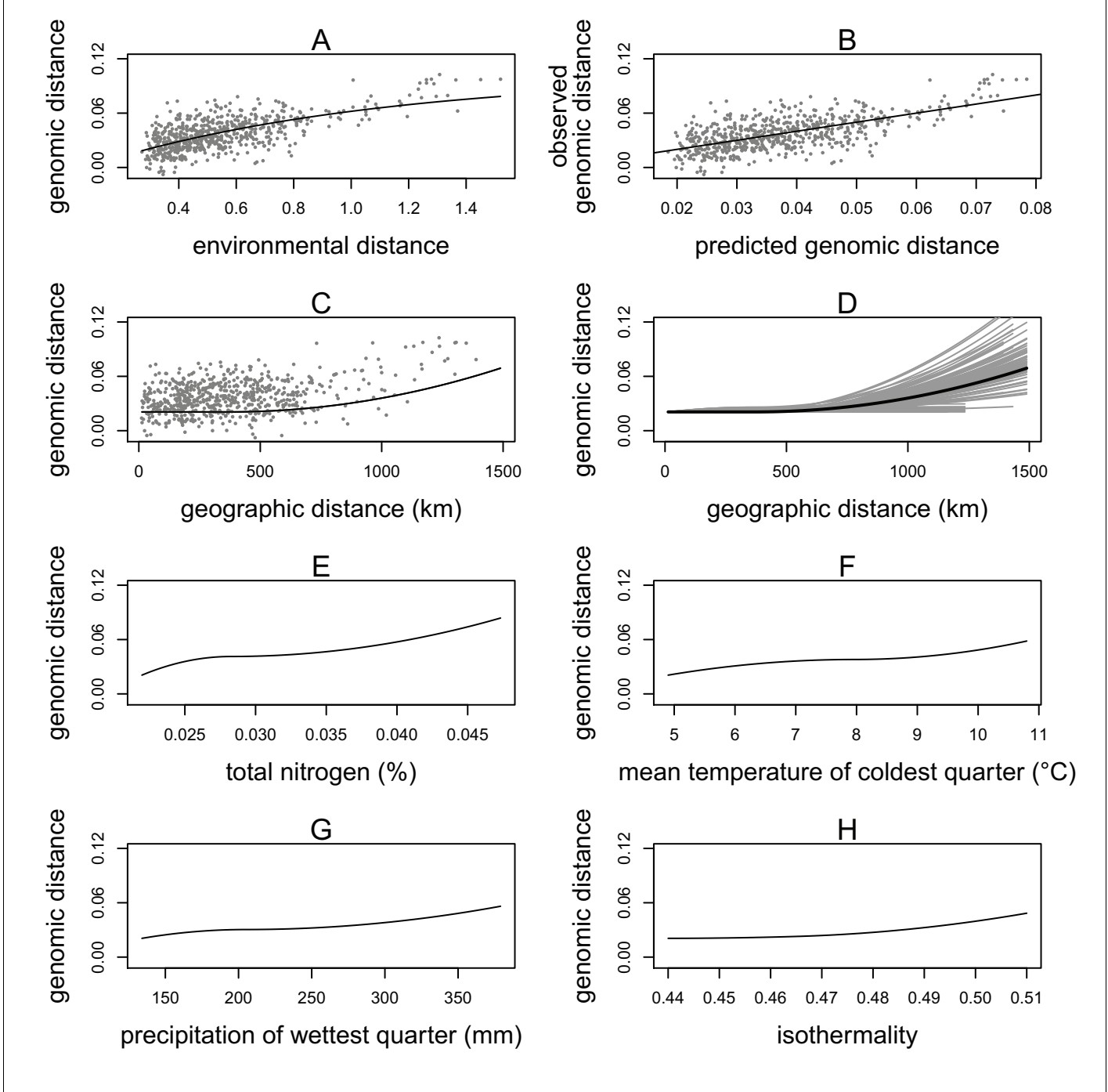

**Figure 2.** Generalized dissimilarity modelling (GDM) results. (**A**) Non-linear relationship between environmental distance and genomic distance. Points are site pairs; the line is the predicted relationship. (**B**) Relationship between predicted genomic distance and observed genomic distance. Points are site pairs; the line indicates where observation and prediction match. (**C**) The geographic spline showing little predicted genomic change between sites less than 500 km apart and increasing genomic variation as geographic distance increases beyond 500 km. Points are site pairs. (**D**) Geographic splines from 100 iterations of sampling 70% of sites. Each grey line is an iteration; the black line is the full model prediction. (**E–H**) Predicted splines showing the estimated relationship between genomic distance and the environmental variable: (**E**) total nitrogen content at 100–200 cm of soil depth, (**F**) mean temperature of the coldest quarter, (**G**) precipitation of the wettest quarter, and (**H**) isothermality.
DOI: https://doi.org/10.7554/eLife.31835.008

(*Figure 2E*). Mean temperature of the coldest quarter was the second strongest predictor, showing changes in genomic distance predicted across the range of temperature values (*Figure 2F*). Precipitation of the wettest quarter was the third strongest environmental predictor, predicting the largest change in genomic distance between 250 and 400 mm of precipitation (*Figure 2G*). Isothermality (mean diurnal range divided by annual temperature range) was the final predictor, predicting the most change in genomic distance at higher values (*Figure 2H*).

To project the final GDM model onto the current environmental landscape, we first delineated the geographic extent of the analysis by defining an *E. melliodora* distribution polygon. We then projected the GDM model onto this region based on the current values of the environmental variables across the landscape. For visualization, the dimensionality was reduced using principal component analysis (PCA) and the first three axes were assigned to RGB colors to represent genomic composition, with similar color for similar predicted genomic composition. The resulting map partitioned the landscape into a number of regions with different predicted genomic compositions, including northern coastal, northern inland, and southern regions (*Figure 3A*). While the biggest differences occurred in regions with few sampling sites, the northern and southern sites have distinct genomic compositions (*Figure 3A*). These projections highlight where environmental filtering of genotypes may have occurred due to differences in selective pressures.

We compared the GDM model projected onto current conditions to the GDM model projected onto 2070 climate predictions as an indication of the amount of genomic change required to keep pace with changes in selective pressures resulting from environmental change ('genomic vulnerability', (*Bay et al., 2018*)) (*Figure 3B*). For the middle north region and the southern areas towards the coast (red in *Figure 3B*), the model predicted more intense natural selection in response to climate change, thus indicating that these areas should be prioritized for assisted migration.

We also used the GDM model to compare the genomic composition under future environmental conditions at a single location to the genomic composition under current climate conditions across the landscape. This comparison is useful for identifying optimal seed sources for restoration sites given climate change scenarios. We demonstrated this utility by selecting two hypothetical reforestation sites and identifying distinct regions that would provide favorable seed sources for each site (*Figure 4*). The analysis for the southern reforestation site identified a large portion of the southern distribution, centered at the reforestation site. For this site it appears that the selected areas are largely a result of the pattern of isolation by distance, in particular the lack of genetic differentiation for long geographical distances. The analysis for the northern reforestation site identified a more limited range of areas across the landscape, although this could be driven in part by a decreased power due to lower sampling intensity in the north. Within 500 km of the site, the analysis identified a core region centered on the reforestation site and small regions along the northern coast. There were a number of areas within 500 km of the site that were not good matches. In addition, a number of more distant areas along the southern coast were also identified, indicating these selected areas are driven more by patterns of isolation by environment than isolation by distance. Overall, the map suggests that there is a lower availability of seed sources to match the northern reforestation site.

These analyses suggest that for seed sourcing in woodland restoration, a model-based approach incorporating genomic variation, geographic distance, and environmental variables would allow for more genetic diversity and enable better matching of the selected genotypes to current and predicted future environmental conditions at the reforestation site.

## Growth experiments

We conducted a climate controlled growth experiment to examine phenotypic variation among sampling sites and assay phenotypic plasticity. We grew seedlings from six sites, with six maternal lines per site, at two different climate regimes (average summer conditions and 5°C hotter than summer conditions). We measured variation in three seedling growth traits: seedling height, total leaf length, and relative height increment. For analysis of seedling height and total leaf length, we analyzed a total of 291 seedlings (from 32 maternal lines representing six sampling sites) that were determined to be well established at the five week measurement. For analysis of the relative height increment, we analyzed a total of 560 seedlings (from all 36 maternal lines) for which we were able to calculate this metric. There were four seedlings that were outliers for the relative height increment. These outliers had little effect on the results of the linear models, so we included them in the final analysis.

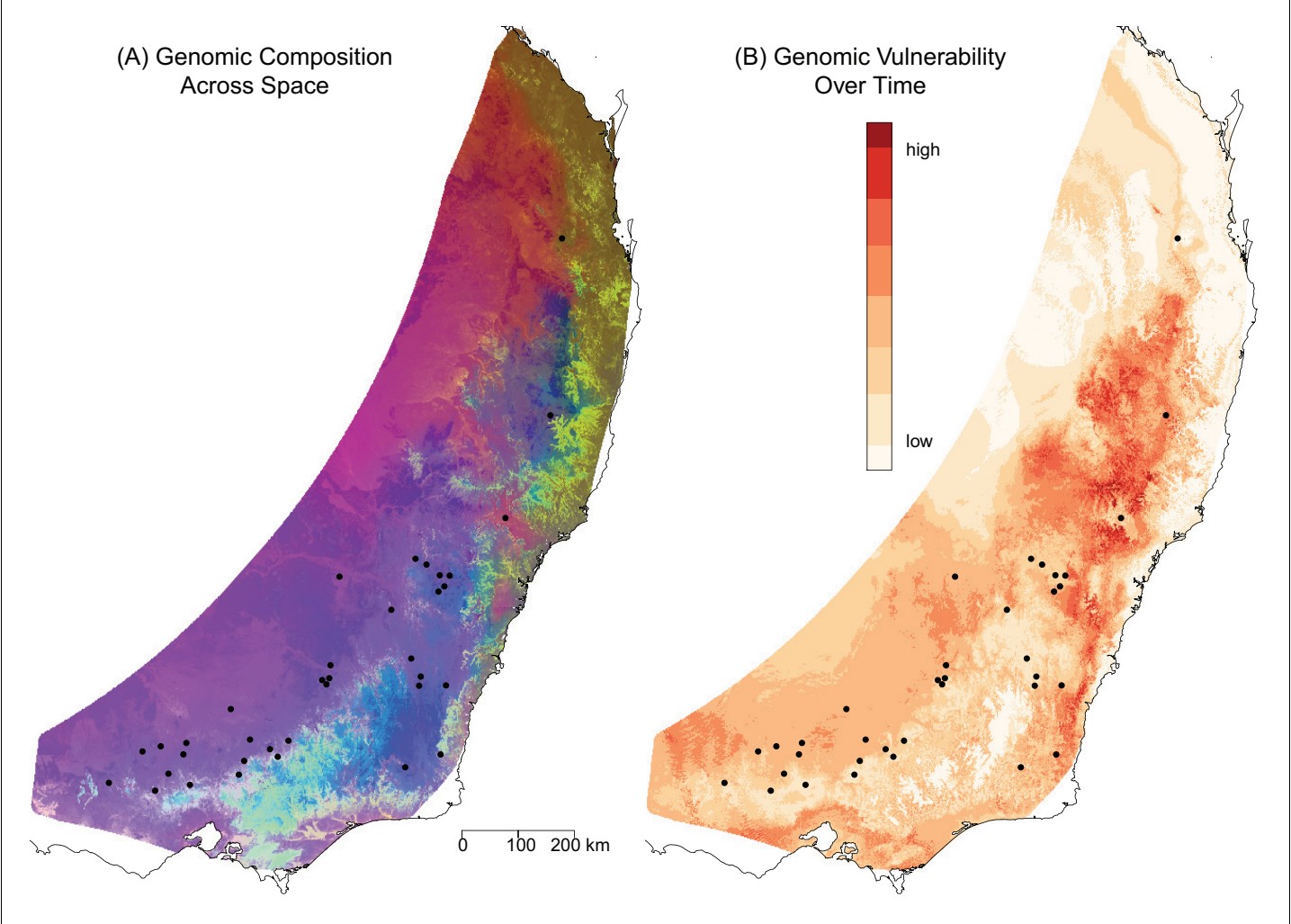

**Figure 3.** Predicted spatial and temporal variation in genomic composition. (**A**) The spatial distribution of predicted genomic variation based on projecting the GDM model onto geography and current environmental conditions. Regions with similar colors are predicted to have similar genomic compositions. (**B**) The predicted genomic vulnerability based on comparing the GDM model projected onto current environmental conditions with the GDM model projected onto predicted environmental conditions for 2070. The higher the difference (darker red), the more genomic change required to track climate between current and future conditions. Black points are sampling sites.

DOI: https://doi.org/10.7554/eLife.31835.009

The models for all three response variables showed that all fixed effects (sampling site, maternal line nested within sampling site, and experimental condition) were statistically significant at the p=0.05 level (*Figure 5* and *Supplementary file 4*). Experimental condition explained a small percentage of the variation (1.2–8.1%), as did sampling site (1.8–17.7%). Maternal line tended to explain a larger amount of variation (10.6–27.6%). However, most of the variation remained unexplained (56.6–71.5%) (*Figure 5* and *Supplementary file 4*). None of the three response variables showed significant variation in phenotypic plasticity across sites (p>0.50 for all maternal line/sampling site by experimental condition interactions) (*Figure 5* and *Supplementary file 5*).

We then conducted an outdoor drought experiment using a subset of seedlings from the chamber experiment. We analyzed 146 seedlings representing 20 maternal lines from five sampling sites. These seedlings were grouped into 73 pairs, with one of each pair assigned to each treatment— well-watered versus drought. We analyzed variation in four response variables: stomatal conductance, leaf length to width ratio, relative chlorophyll content (SPAD index), and specific leaf area (SLA, leaf area divided by dry mass).

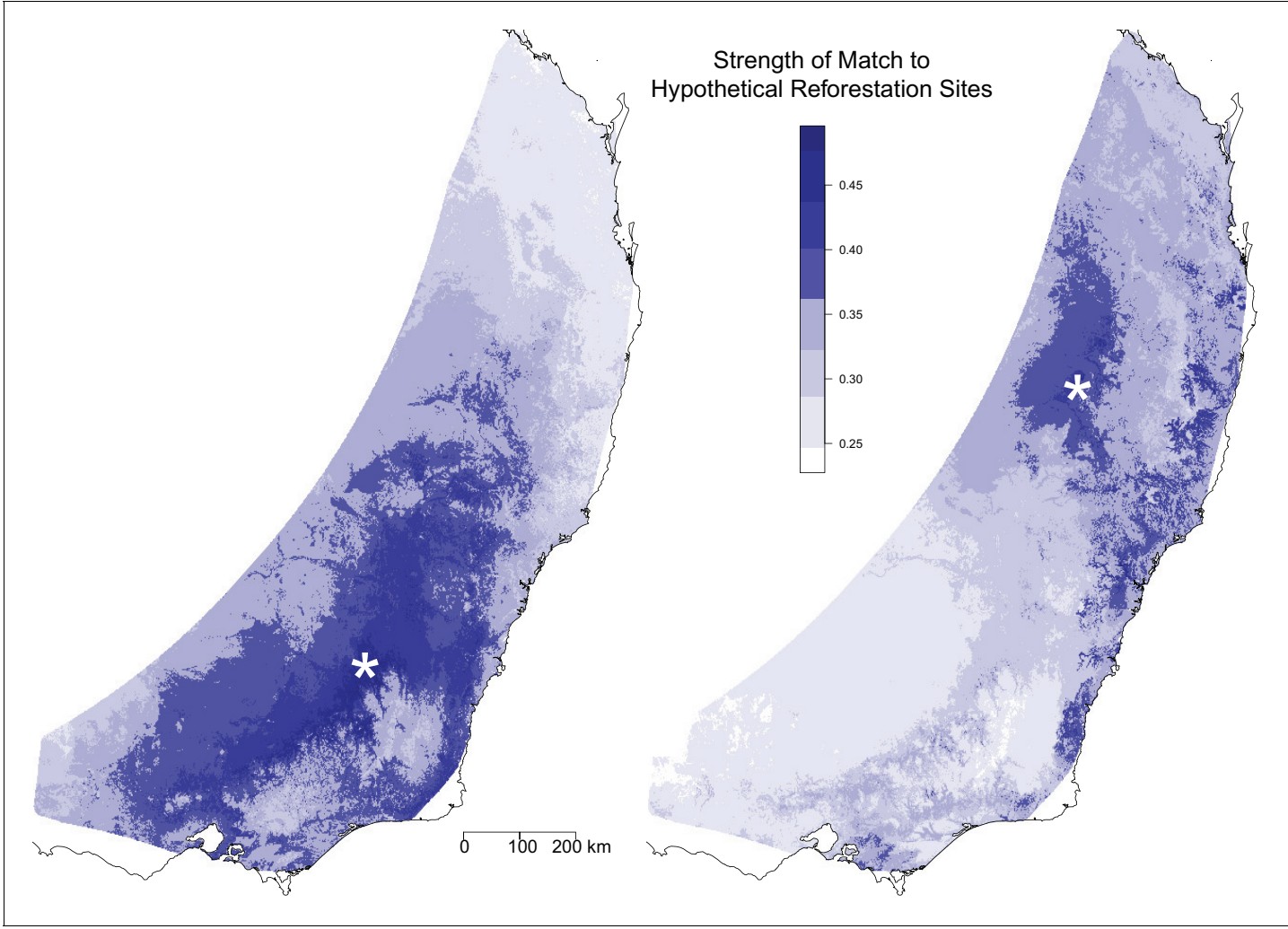

**Figure 4.** Optimal seed sourcing locations for hypothetical reforestation sites. The predicted genomic similarity of hypothetical reforestation sites (indicated by white asterisks) to potential seed sourcing locations under a climate change scenario for 2070. Dark blue areas indicate seed sourcing areas predicted to best match future conditions at the hypothetical reforestation sites; white and light blue areas indicate areas of potential genomic mismatch.

DOI: https://doi.org/10.7554/eLife.31835.010

The drought-treated seedlings had significantly lower stomatal conductance rates than the well-watered ones, indicating that the seedlings were affected by the watering treatment (p<0.00001) (*Figure 6* and *Supplementary file 6*). Treatment explained most of the variation in stomatal conductance (62.3%), while maternal line and sampling site explained only a small amount of variation (5.8% and 0.9% respectively). For the remaining three response variables (leaf length to width ratio, SPAD, and SLA), much of the variation was unexplained (40.5%–70%). Treatment was not statistically significant and explained little to no variation (0.0–4.4%). Sampling site and maternal line were statistically significant in the linear models at the p=0.05 level and explained some variation (6.7–21.2%) (*Figure 6* and *Supplementary file 6*). Smaller, thicker leaves, and thus lower SLA values, were expected for drought-treated seedlings and for seedlings grown from seed collected from drier areas. Consistent with this expectation, the seedlings subjected to drought conditions showed lower SLA values. However, seedlings from drier sampling sites (D and T3) showed higher SLA values than more mesic sites (B, G, and 11), contrary to expectation (*Figure 6*). None of the four response variables showed significant variation in phenotypic plasticity across sites (p>0.13 for all maternal line/sampling site by experimental condition interactions) (*Figure 6* and *Supplementary file 7*).

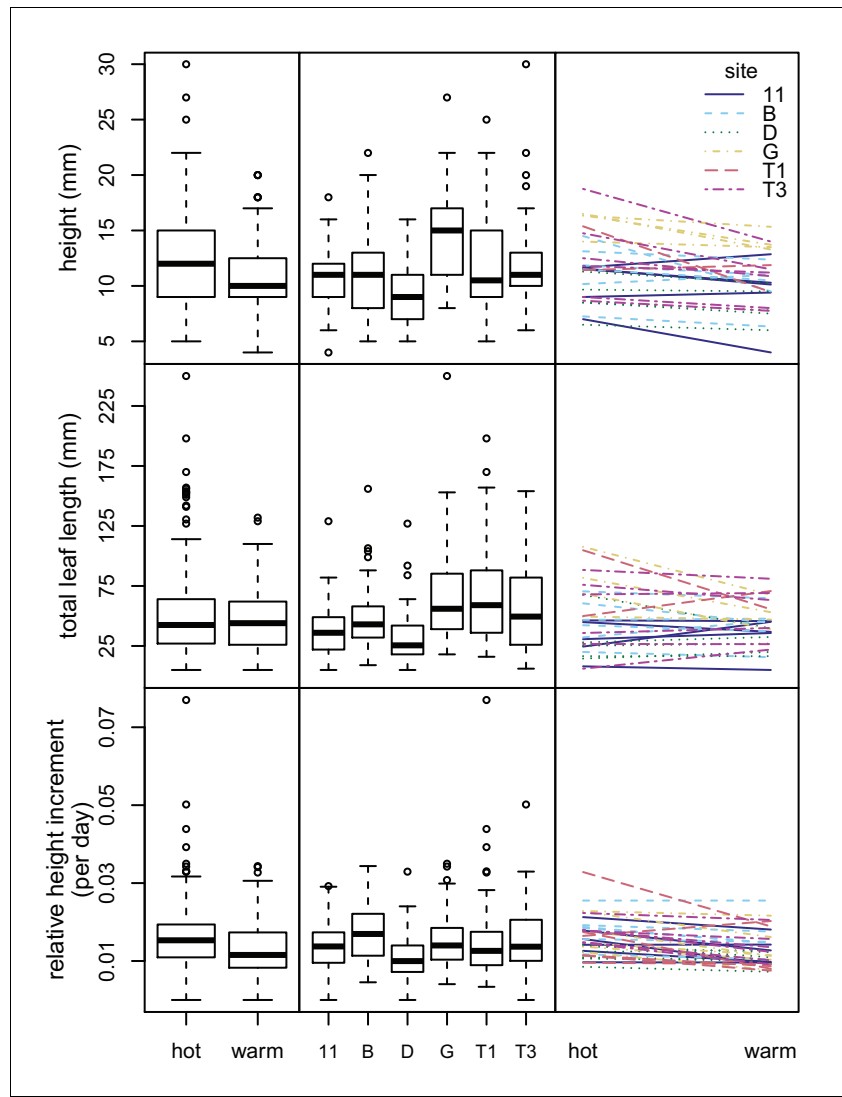

**Figure 5.** Variation in seedling growth in chamber experiment. Box plots showing variation between experimental conditions (left) and sampling sites (center) for three seedling growth traits. Plots showing interactions between seedling growth traits and experimental conditions (right). Each line represents a maternal line, with color and line type indicating the sampling site.

DOI: https://doi.org/10.7554/eLife.31835.011

Both our climate controlled growth experiment and our outdoor drought experiment found high levels of phenotypic variation in all measured traits. While most of the variation remained unexplained, sampling site explained a small, but statistically significant, amount of the variation. We determined whether phenotypic divergence between sites could be due to local selection using a $Q_{st}$-$F_{st}$ analysis (*Gilbert and Whitlock, 2015*; *Leinonen et al., 2013*). We estimated $Q_{st}$ for each trait under each experimental condition and compared these values to the genome-wide distribution of $F_{st}$ values (*Supplementary file 8*). $Q_{st}$ and $F_{st}$ were not significantly different, indicating that phenotypic differences between sites could be a result of genetic drift alone. While not statistically significant, seedling height did show differences between $Q_{st}$ and $F_{st}$ in both hot ($Q_{st}$-$F_{st}$ = 0.33, p=0.11) and warm ($Q_{st}$-$F_{st}$ = 0.24, p=0.14) chambers. This indicates that local selection could be driving the divergence in height between sites, but our analysis lacked statistical power due to small sample sizes.

In addition to measuring growth traits, we also examined the shape of the leaves of seedlings from the drought experiment. We noted substantial variation in leaf shape, both among sites and

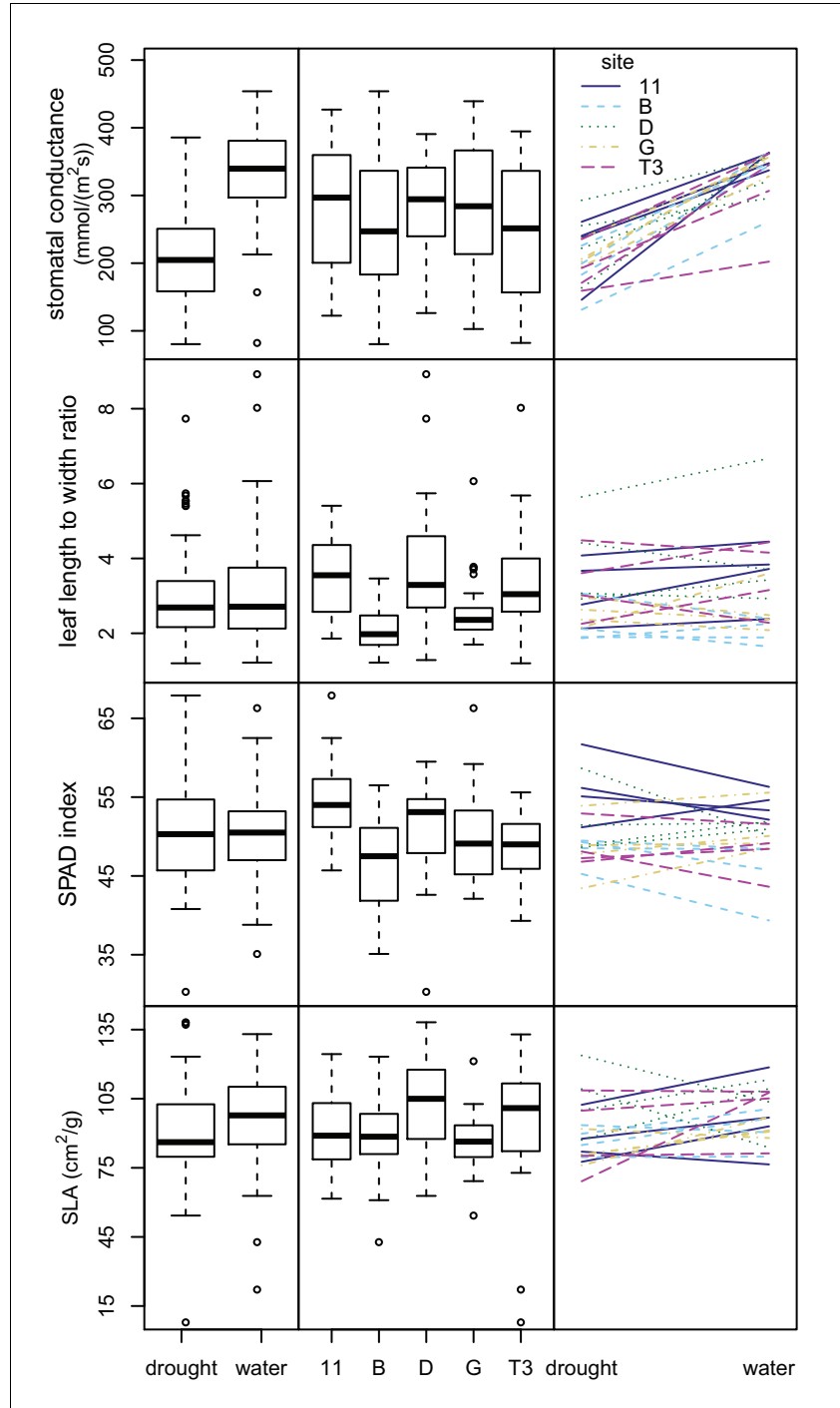

**Figure 6.** Variation in leaf traits in drought experiment. Box plots showing variation between water treatments (left) and sampling sites (center) for four leaf traits. Plots showing interactions between leaf traits and water treatments (right). Each line represents a maternal line, with color and line type indicating the sampling site.
DOI: https://doi.org/10.7554/eLife.31835.012

within sites (*Figure 7*). The remarkable amount of phenotypic variation in the seedlings is consistent with the high levels of genomic variation measured.

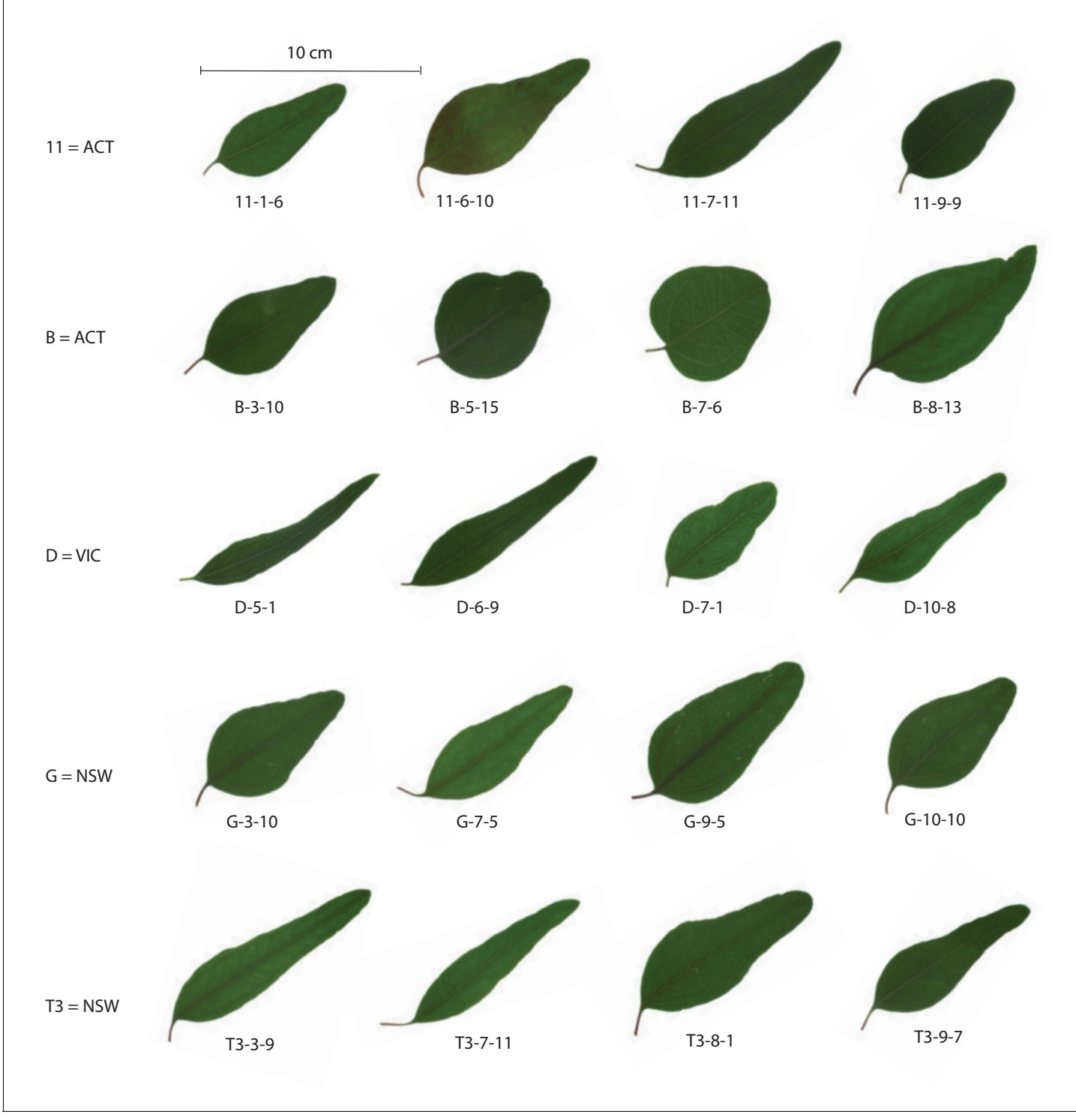

**Figure 7.** Variation in leaf shape. One representative leaf from each maternal line in the drought experiment. Each row shows a single sampling site, identified by site ID and state location (ACT = Australian Capital Territory, VIC = Victoria, NSW = New South Wales). Each leaf is identified by its sampling site, maternal line, and replicate number).

DOI: https://doi.org/10.7554/eLife.31835.013

## Discussion

*Eucalyptus melliodora* is a foundation species in a critically endangered woodland community that now occupies a fraction of its former distribution and is the subject of restoration projects across its native range. Our examination of the distribution of genomic and phenotypic variation across the range of this species provides valuable information for sourcing seed for restoration, including empirically defining local provenances and matching genotypes to predicted future environmental conditions.

We found little genomic divergence between sampling sites (mean $F_{st}$ = 0.04), which is consistent with microsatellite analysis of this species ($F_{st}$ = 0.03, (*Broadhurst et al., 2018*)) and population genetic analyses of other tree species (*E. camaldulensis*, $F_{st}$ = 0.05, 0.08, (*Butcher et al., 2009*); *E. globulus*, $F_{st}$ = 0.08, (*Jones et al., 2002*); *Corymbia calophylla*, $F_{st}$ = 0.03, (*Sampson et al., 2018*); *Pinus taeda*, $F_{st}$ = 0.04, (*Eckert et al., 2010*); *Quercus robur*, $F_{st}$ = 0.07, (*Vakkari et al., 2006*); *Quercus engelmannii*, $F_{st}$ = 0.04, (*Ortego et al., 2012*); *Populus tremuloides*, $F_{st}$ = 0.03, (*Wyman et al., 2003*)).

Examining the relationship between genomic and geographic distance, we are able to empirically define 'local' in this species to be on the order of 500 km, which is substantially farther than the current practice. These results mean restoration projects can and should source seed more broadly across the landscape, with limited risk of mixing highly evolutionary diverged material. In a highly fragmented landscape this will increase the number of favorable source sites, enabling the collection of higher quality seed with increased genetic diversity (*Broadhurst et al., 2008*). Incorporating more naturally occurring genomic variation can increase the adaptive potential of the restored populations by providing the substrate for adaptation to rapidly changing environmental conditions.

In addition to isolation by distance, our model identified soil nitrogen, temperature of the coldest quarter, precipitation of the wettest quarter, and isothermality as significant environmental drivers of genome-wide patterns of variation across the landscape. Of these variables, the climate variables are predicted to change rapidly over time. Change in soil nitrogen content might occur over longer time scales, but it is difficult to forecast due to complex biotic feedbacks (*Brevik, 2013*). This suggests that optimal seed sourcing will need to balance the tracking of rapidly changing climate variables with the need to account for variables that are more stable due to their dependence on stable features of geology, topography, or hydrology. The different time scales also highlight the important concern that key environmental variables may become uncoupled, resulting in less than ideal conditions for this species across the landscape.

Previous niche modelling of *E. melliodora* examined environmental drivers of the distribution of the species (*Broadhurst et al., 2018*). Similar to our analysis, that analysis also found temperature and precipitation variables to be important, but the exact bioclimatic variables identified did not overlap. This is not unexpected given that niche modelling identifies drivers that define the environmental tolerance of the species, while the analysis presented here identifies drivers for genomic variation within the species.

Many studies of within-species genetic variation in trees find temperature and precipitation variables to be the most important drivers (*Aitken et al., 2008*); however, the exact variables vary and other variables are often found to play important roles. A quantitative genetics study of Eucalyptus *delegatensis* in Australia found that the variables that contributed most to the adaptive variability of the species were related to solar radiation (*Garnier-Géré and Ades, 2001*), which was not assessed in our study. Additionally, they found that the variability of temperature and rainfall played an important role (*Garnier-Géré and Ades, 2001*). One of our top predictors was isothermality, which is a composite variable of temperature ranges.

A genetic study of ecologically relevant loci in 13 alpine plant species in the European Alps found that, after accounting for broad spatial patterns, temperature and/or precipitation variables were the primary drivers of genetic variation in all but one species (*Manel et al., 2012*). In contrast, a genetic study of putatively neutral loci in three tree species in Central America found different drivers in different species (*Poelchau and Hamrick, 2012*). In one species, an integrated environmental measure, incorporating temperature and precipitation, was the primary driver; in the second species the primary driver was geographic distance; in the third species the results were ambiguous (*Poelchau and Hamrick, 2012*). This indicates that environmental drivers of within-species genetic diversity are likely to be somewhat species specific.

The focus of this study was the whole-genome population structure that reflects historical adaptation, gene flow, and demography. Analyses of individual genes was beyond the scope of this study due to the low resolution GBS genotyping and the limited extent of linkage disequilibrium in *Eucalyptus* (*Silva-Junior and Grattapaglia, 2015*; *Thumma et al., 2005*). However, our results demonstrate a lack of strong population structure, indicating that using whole genome sequencing to identify adaptive alleles is feasible in this species. For instance, the $Q_{st}$-$F_{st}$ analysis indicates the possibility of local adaptation for seedling height and a future study could identify the adaptive loci underlying plant height by targeting sampling sites segregating for this trait. Specific alleles that potentially confer increased fitness in the face of a rapidly changing climate would be useful targets for restoration projects.

Our analyses of phenotypic variation found no site-specific variation in phenotypic plasticity that would enable us to identify provenances better able to cope with rapid environmental change. However, plasticity is trait-specific, so traits that are hypothesized to be important for establishment and survival should continue to be investigated because they may provide valuable information for restoration projects. Importantly, our growth experiments support the results of the genomic analyses, showing the remarkable extent of variation both among sites and within sites, further supporting our recommendation that seed sources incorporate the high level of variation that occurs naturally in *E. melliodora*.

The results of this study are promising for the future of *E. melliodora* across its native distribution. We found high genomic and phenotypic diversity within sites, as well as shared across the range. This naturally occurring variation can provide a basis for adaptation to rapidly changing environments and it should be incorporated into restoration projects through strategic seed sourcing. It is important to note that our genomic analyses were based on mature trees that predate extensive land clearing for agriculture. It remains to be determined whether human modifications of the landscape have disrupted the historical patterns of gene flow, resulting in more fragmented and inbred populations. Genomic analyses of seedlings or saplings at these sites may show different results, although our phenotypic studies using seedlings produced results concordant with our genomic analyses.

Our landscape genomic model can guide seed selection by empirically defining local provenances and identifying variation suitable for predicted future climates. This understanding of the relationship between environmental and genomic variation can be combined with other types of information, such as basic biological knowledge of the ecological community and best agronomic practices in restoration, to establish foundation species and ecosystems with the highest probability of success in rapidly changing environments.

## Materials and methods

### Sample collection

We obtained *E. melliodora* leaf samples from mature trees at 39 sampling sites—38 sites across the species' native range and a single site in Western Australia, well outside the species' natural distribution. We collected samples through a community science project described in *Broadhurst et al., 2018* (*Supplementary file 1*). From each site, a citizen scientist collected leaf samples from up to 30 trees, put the samples in silica gel for drying, and shipped them to CSIRO for processing. In addition to leaf material, they also collected seeds from the sampled trees when available.

### Genotyping by sequencing

We selected 3 to 10 trees per sampling site for sequencing and we processed each of the seven trees from Western Australia twice, using different leaves from the same tree to serve as technical replicates. No power analysis was used to determine sample size during the design of the study. Sample size was determined based on our experience and judgment, with consideration of the availability of samples. We sequenced these 379 samples using a modified Genotyping-By-Sequencing (GBS) protocol (*Elshire et al., 2011*). Briefly, we extracted genomic DNA from approximately 50 mg of leaf tissue using the Qiagen DNeasy Plant 96 Kit, digested with PstI for genome complexity reduction, and ligated with a uniquely barcoded sequencing adapter pair. We then individually PCR amplified each sample to avoid sample bias. We pooled samples in equimolar concentrations and

extracted library amplicons between 350 and 600 bp from an agarose gel. We sequenced the library pool on an Illumina HiSeq2500 using a 101 bp paired-end protocol at the Biomolecular Resource Facility at the Australian National University, generating almost 260 million read pairs.

We checked the quality of the raw sequencing reads with FastQC (v0.10.1, [*Andrews, 2012*]). We used AXE (v0.2.6, [*Murray and Borevitz, 2017a*]) to demultiplex the sequencing reads according to each sample's unique combinatorial barcode and were unable to assign 11% of read pairs to a sample. We used *trimit* from libqcpp (v0.2.5, [*Murray and Borevitz, 2017b*]) to clean the reads for each sample, using default parameters, except q = 20. This involved removing adapter contamination due to read-through of small fragments (20% of read pairs) and merging overlapping pairs (49% of read pairs), with both steps using algorithms based on a global alignment of the read pair. We also used *trimit* for sliding window quality trimming (11% of reads). We then used a custom script to remove sequencing reads that did not begin with the expected restriction site sequence (16% of reads). We aligned sequencing reads to the *E. grandis* reference genome (v2.0, [*Bartholomé et al., 2015*; *JGI, 2015*; *Myburg et al., 2014*]), including all nuclear, chloroplast, mitochondrial, and ribosomal scaffolds, but used only nuclear scaffolds for downstream analyses. We aligned reads using bwa-mem (v0.7.5a-r405, [*Li, 2013*]), as paired reads (-p) and treating shorter split hits as secondary alignments (-M), with 88% of reads successfully mapped. We used GATK's *HaplotypeCaller* in GVCF mode (v3.6–0-g89b7209, [*McKenna et al., 2010*]) to call variants for each sample with heterozygosity (-hets) increased to 0.005, indel heterozygosity (-indelHeterozygosity) increased to 0.0005, and the minimum number of reads sharing the same alignment start (-minReadsPerAlignStart) decreased to 4.

We used GATK's *GenotypeGVCFs* (v3.6–0-g89b7209, [*McKenna et al., 2010*]) for a preliminary round of joint genotyping across all samples based on the individual variant calls and again increasing the heterozygosity (-hets) to 0.005 and the indel heterozygosity (-indelHeterozygosity) to 0.0005. For basic filtering, we used GATK to remove variants that were indels, had no variation relative to the reference, were non-biallelic SNPs, had QD < 2.0 ('variant call confidence normalized by depth of sample reads supporting a variant'), MQ > 40.0 ('Root Mean Square of the mapping quality of reads across all samples'), or MQRankSum < -12.5 ('Rank Sum Test for mapping qualities of REF versus ALT reads'). We removed samples with more than 60% missing data and SNPs with more than 80% missing data. We examined the genomic distance between samples to verify that technical replicates clustered closely together. We used a preliminary PCoA, based on genomic distance between samples, to identify outlier samples. We removed outlier samples and poorly sequenced samples from the dataset for final genotyping and all downstream analyses.

We reran GATK's joint genotyping on the final sample set. We again used GATK to remove variants that were indels, SNPs with no variation relative to the reference, and non-biallelic SNPs. We determined final filtering thresholds by examining parameter distributions. A locus was retained for subsequent analysis if ExcessHet < 13.0 ('phred-scaled p-value for exact test of excess heterozygosity'), -0.3 < InbreedingCoeff < 0.3 ('likelihood-based test for the inbreeding among samples'), MQ > 15.0 ('Root Mean Square of the mapping quality of reads across all samples'), -10.0 < MQRankSum < 10.0 ('Rank Sum Test for mapping qualities of REF versus ALT reads'), and QD > 8.0 ('variant call confidence normalized by depth of sample reads supporting a variant'). We ran a second preliminary PCoA analysis to identify additional outlier samples. Finally, we used VCFtools (v0.1.12b, [*Danecek et al., 2011*]) to remove SNPs with greater than 60% missing data and thin the SNPs so that none were closer than 300 bp.

## Genomic analyses

To examine the genomic structure of *E. melliodora* and how it is influenced by geography, we conducted individual-based analyses. For these analyses, we converted the final genotypic data (a vcf file) to a sample-by-SNP matrix and imported it into a *genind* object (R adegenet v2.0.1, [*Jombart, 2008*]). We calculated the pairwise genomic distances between individuals using a euclidean distance in *dist* (R stats v3.1.2, [*R Core Team, 2015*]). To visualize the genomic distance among samples, we ran a PCoA using *dudi.pco* (R ade4 v1.7–4, [*Dray and Dufour, 2007*]). We plotted the first two PCoA axes, with samples colored in a rainbow gradient based on sample latitude. We calculated the linear regression and correlation between latitude and the first PCoA axis using *lm* (R stats 3.1.2, [*R Core Team, 2015*]). We calculated the geographic distance between samples based on their GPS coordinates using *earth.dist* (R fossil v0.3.7, [*Vavrek, 2011*]). We used a *mantel* test (R vegan v2.4–0,

[*Oksanen et al., 2016*]), which examines the correlation between two distance matrices, to quantify the linear relationship between the genomic distance between individuals and the natural logarithm of the geographic distance.

We then conducted site-based analyses. To estimate within-site genomic diversity, for each sampling site we calculated the number of alleles and the expected heterozygosity using *summary* and *Hs* (R adegenet v2.0.1, [*Jombart, 2008*]). We used the sample-by-SNP matrix to calculate pairwise $F_{st}$ (*Weir and Cockerham, 1984*) using *pairwise.WCfst* (R hierfstat v0.04–22, [*Goudet and Jombart, 2015*]). We ran a sampling-site level PCoA on the pairwise $F_{st}$ matrix using *dudi.pco* (R ade4 v1.7–4, [*Dray and Dufour, 2007*]) and calculated the percent of variation explained for each PCoA axis.

To examine the role that environmental factors played in driving the genomic structure across the landscape, we used Generalized Dissimilarity Modelling (GDM), which uses matrix regression to estimate the non-linear relationship between genomic distance and environmental distance (*Ferrier et al., 2007*; *Fitzpatrick and Keller, 2015*; *Thomassen et al., 2011*). We then used this model to predict the distribution of genomic variation across the landscape under current environmental conditions, as well as predicted future conditions.

We obtained environmental data for the GDM from climate, elevation, soil, and landscape raster layers. Climate variables included 19 bioclimatic variables for the current time period (1960–1990), at 30 arc second resolution (*WorldClim, 2016b*). Elevation was from a digital elevation model aggregated from 90 m resolution (*CGIAR-CSI, 2016*). Soil data included available water capacity, total nitrogen, and total phosphorus sampled at the surface (0–5 cm) and at depth (100–200 cm) (*CSIRO, 2016*). Landscape data included the Prescott Index (a measure of water balance) and MrVBF (a topographic index) (*CSIRO, 2016*). For future predictions, we used the 19 bioclimatic variables predicted for 2070 at 30 arc second resolution based on GCM MIROC5 for representative concentration pathway 8.5 (*WorldClim, 2016a*), which is a greenhouse gas concentration trajectory showing continual increase in emissions over time. We determined the values for each variable at each sampling site based on GPS coordinates and used those values to calculate the environmental distances between sites.

To determine the genomic distances between sampling sites for the GDM, we scaled the $F_{st}$ matrix to between 0 and 1 by subtracting the minimum value and then dividing by the maximum value. We generated the GDM model using *gdm* (R gdm v1.2.3, [*Manion et al., 2016*]) with the scaled $F_{st}$ matrix, geographic distances between sites, and environmental distances for the 28 variables for the current time period. Initially, we generated a GDM model for each environmental variable separately and excluded variables from further analysis if the deviance explained by the model was less than 5%. For the remaining variables, we calculated Pearson's correlation for site values between pairwise sets of variables. If a pair of variables had a correlation greater than 60%, we excluded the variable with the lowest explanatory power from subsequent analysis. We conducted permutation testing using *gdm.varImp* (R gdm v1.2.3, [*Manion et al., 2016*]) with 1000 permutations to determine the explanatory power and statistical significance of the remaining variables and to excluded additional inconsequential variables. We generated a final GDM model with the remaining environmental variables.

We cross validated the GDM model using a random 70% of the spatial sampling sites as training data and the remaining 30% of sites as test data and ran 1000 resampled iterations. We used the GDM models from the training sites to predict the genomic dissimilarity between the test sites and used Pearson's correlation to compare the predicted values to the observed values. To test the robustness of the geographic prediction from the GDM model, we visualized the geographic splines from 100 of these GDM models.

To project the final GDM model onto the current environmental landscape, we first delineated the geographic extent of the analysis by defining an *E. melliodora* distribution polygon. We downloaded 14,977 *E. melliodora* occurrence records from the Atlas of Living Australia (*ALA, 2016*), of which we removed 189 because they were well outside the expected distribution or were sparse records on the distribution margin. We generated the polygon using *ahull* (R alphahull v2.1, [*Pateiro-López and Rodríguez-Casal, 2010*]), with alpha = 15 and *gBuffer* (R rgeos v0.3–21, [*Bivand and Rundel, 2016*]), with a 20 km buffer. We then transformed the environmental rasters based on the model splines (*gdm.transform*), performed a PCA of the transformed layers (*prcomp* R stats v3.1.2, [*R Core Team, 2015*]), and predicted across space (*predict*). We visualized the result by graphing the first three components of a PCA using a red-green-blue plot (*Fitzpatrick and Keller,*

*2015*). We also projected the model onto a predicted future environmental landscape with the same procedure, except we replaced the current bioclimatic rasters with the future ones for 2070 that were predicted under a high $CO_2$ emission scenario. We calculated 'genomic vulnerability' (*Bay et al., 2018*), which is the amount genomic change required to track environmental change over time, using the *predict* function with time = TRUE.

We examined the implications of the GDM model for seed sourcing decisions by selecting two hypothetical reforestation sites. We compared predicted future GDM values at these two hypothetical reforestation sites to current climate GDM values across the landscape of potential seed sources. This enabled us to generate a map of the predicted genomic similarity of potential seed sources to the hypothetical reforestation sites under climate change.

## Growth experiments

To examine the effect of provenance and environment on phenotype, we conducted experiments in climate controlled growth chambers under two different climate regimes. No power analysis was used to determine sample size during the design of the experiment. Sample size was determined based on our experience and judgment, with consideration of the availability of seed and space in the growth chambers. We selected six sites (11, B, D, G, T1, T3; asterisks in *Figure 1A*) and six maternal lines per site that had sufficient seed. For each of the 36 maternal lines, we grew a minimum of 64 replicate seedlings, with four seeds planted per pot (6.5 cm x 6.5 cm x 20 cm pots with soil that was 80% Martin's mix and 20% sand). We germinated seeds in climate controlled chambers with 12 hr of light at 25°C and 12 hr of dark at 15°C. We set lights to mimic summer morning light (photosynthetic photon flux 370 nm = 82, 400 nm = 83, 420 nm = 78, 450 nm = 37, 530 nm = 31, 620 nm = 72, 660 nm = 28, 735 nm = 34, 850 nm = 89, 6500 K = 94 µmol/m$^2$/s). We watered all seeds twice daily to keep the soil moist. We culled to one seedling per pot 12–14 days after planting.

Three weeks after germination, we sorted seedlings into treatment chambers using a randomized block design based on maternal line. In each of the two climate chambers, we grew eight or nine replicate seedlings from each maternal line. Climate conditions were determined with SolarCalc (*Spokas and Forcella, 2006*) to mimic average summer conditions (sampling site 11) and hotter conditions (5°C temperature increase; sampling site T3). We ran the experimental conditions for 12–14 weeks and took phenotypic measurements at five time points:1, 2, 3, 5, and 11 weeks after the experimental treatment began. Measurements included seedling height, number of leaves, and total leaf length.

For the analysis of seedling height and total leaf length, we used the measurements at five weeks after the experimental treatment began and used only seedlings that were determined to be well established at that time. We also calculated a relative height increment for each seedling by determining the last measurement when the seedling had two or fewer leaves and the first measurement with eight or more leaves. The relative height increment is the difference between the natural logarithm of the two height measurements, divided by the difference in time.

We investigated phenotypic plasticity by examining interaction plots between maternal line and experimental condition for three response variables: seedling height, total leaf length, and relative height increment. We statistically tested for an interaction between sampling site/maternal line and experimental condition with linear mixed-effect models using *lmer* (R lme4 v1.1–10, [*Bates et al., 2015*]) for each of the three response variables. Due to a lack of power to consider maternal line nested within sampling site, we ran two models for each response variable—one with maternal line and one with sampling site. These models included the experimental condition, sampling site or maternal line, and their interactions as fixed effects. We included germination chamber and block as random effects. We visually identified four outliers with a relative height increment over 0.035. We ran the models with and without outliers to determine if they affected the results.

We visualized the distribution of values for the three response variables across the six sampling sites using box plots. We quantified the distribution of phenotypic variation with linear mixed-effect models using *lmer* (R lme4 v1.1–10, [*Bates et al., 2015*]). For each of the three response variables, the model included maternal line nested within sampling site and experimental condition as main effects, with no interaction term, and germination chamber and block as random effects.

After completion of the chamber experiment, we conducted an outdoor covered drought experiment on the 16 week old seedlings. No power analysis was used to determine sample size during

the design of the experiment. Sample size was determined based on our experience and judgment, with consideration of the availability of space in the covered growth facility. We selected 160 seedlings from five sampling sites, with four maternal lines per site. We paired each seedling with a seedling of similar size from the same maternal line and treatment chamber. We randomly assigned each seedling of the pair to a different treatment group. We transplanted the seedlings to PVC tubes (9 cm diameter x 50 cm height with sand, perlite, and slow release osmocote) and kept them well watered for seven weeks, allowing them to acclimate to the outdoor conditions. Then we imposed two treatments: well-watered and drought. For the well-watered treatment, we watered the seedlings to saturation as needed (between three times per week and twice per day, depending on the weather). For the drought treatment, we watered as necessary to reach (but not exceed) 50% saturation.

We measured leaf traits on each seedling three weeks after the initiation of treatment. We measured stomatal conductance with a porometer (SC-1 Leaf Porometer by Decagon Devices) and determined that water stress was induced in the drought-treated seedlings. We determined the leaf length to width ratio from a scan of the most recent fully expanded leaf from each seedling using image analysis software (WD3 WinDIAS Leaf Image Analysis System by Delta-T Devices). This leaf was initiated prior to the start of treatment, but expanded while under treatment conditions. We took additional measurements two months after the initiation of treatment. We used a chlorophyll meter (SPAD – 502 by Konica Minolta) to determine the SPAD index, which measures relative chlorophyll content; reduction in SPAD index would indicate detrimental effects of water limitation. We calculated specific leaf area (SLA, leaf area divided by dry mass) by scanning a single leaf from each seedling to determine the leaf area (WD3 WinDIAS Leaf Image Analysis System by Delta-T Devices) and weighing oven dried leaves. For analysis, we excluded data for seedlings that died during the experiment. We also excluded the experimental treatment partner of any dead seedlings.

We visualized phenotypic plasticity by examining interaction plots between maternal line and experimental condition for four response variables: stomatal conductance, leaf length to width ratio, SPAD index, and SLA. We statistically tested for an interaction between sampling site/maternal line and experimental condition with linear mixed-effect models using *lmer* (R lme4 v1.1–10, [*Bates et al., 2015*]) for each of the four response variables. Due to a lack of power to consider maternal line nested within sampling site, we ran two models for each response variable—one with maternal line and one with sampling site. These models included the experimental condition, sampling site or maternal line, and their interactions as fixed effects. We included block and sample pairings as random effects.

We visualized the distribution of values for the four response variables across the five sampling sites using box plots. We quantified the distribution of phenotypic variation with linear mixed-effect models using *lmer* (R lme4 v1.1–10, [*Bates et al., 2015*]). For each of the four response variables, the model included maternal line nested within sampling site and experimental condition as main effects, with no interaction term, and block and sample pairings as random effects. Due to a lack of power, the p-value for the sampling site term was determined from a model without maternal line.

We examined local adaptation using a $Q_{st}$-$F_{st}$ analysis (R QstFstComp v0.2, [*Gilbert and Whitlock, 2015*]) for each phenotypic trait measured under each experimental condition. For each comparison, we estimated $Q_{st}$ under the model for offspring related as half-siblings through shared mothers and compared that value to the distribution of $F_{st}$ values for the sampling sites included in the experiment. Statistical significance was determined based on the predicted null distribution of $Q_{st}$-$F_{st}$ using 10,000 simulation replicates.

## Data access

GBS sequencing reads are available at the NCBI Sequence Read Archive (SRA) (http://www.ncbi.nlm.nih.gov/sra) under BioProject PRJNA413429. Growth experiment data and scripts for genomic and phenotypic analyses are available at https://github.com/LaMariposa/emelliodora (*Supple, 2018*; copy archived at https://github.com/elifesciences-publications/emelliodora).

## Acknowledgements

We thank the ANU Bioinformatics Consultancy for computational support and advice, the ANU Statistical Consulting Unit for statistical advice, the Centre for Biodiversity Analysis for advice on GDM

modelling, and the ANU NCRIS Plant Growth Facility and RSB Plant Services for assistance with growth experiments.

## Additional information

### Funding

| Funder | Grant reference number | Author |
| --- | --- | --- |
| Australian Research Council | Linkage Grant LP130100455 | Jason G Bragg<br>Linda M Broadhurst<br>Adrienne B Nicotra<br>Margaret Byrne<br>Justin O Borevitz |

The funders had no role in study design, data collection and interpretation, or the decision to submit the work for publication.

### Author contributions
Megan Ann Supple, Conceptualization, Data curation, Formal analysis, Investigation, Visualization, Methodology, Writing—original draft, Writing—review and editing; Jason G Bragg, Adrienne B Nicotra, Margaret Byrne, Justin O Borevitz, Conceptualization, Formal analysis, Funding acquisition, Methodology, Writing—review and editing; Linda M Broadhurst, Conceptualization, Resources, Formal analysis, Funding acquisition, Methodology, Writing—review and editing; Rose L Andrew, Conceptualization, Formal analysis, Methodology, Writing—review and editing; Abigail Widdup, Nicola C Aitken, Investigation, Writing—review and editing

### Author ORCIDs
Megan Ann Supple (ID) http://orcid.org/0000-0002-0204-7852
Margaret Byrne (ID) http://orcid.org/0000-0002-7197-5409

### Decision letter and Author response
Decision letter https://doi.org/10.7554/eLife.31835.026
Author response https://doi.org/10.7554/eLife.31835.027

## Additional files

### Supplementary files
• Supplementary file 1. Table of *E. melliodora* sampling information.
DOI: https://doi.org/10.7554/eLife.31835.014

• Supplementary file 2. Table of Pairwise $F_{st}$.
DOI: https://doi.org/10.7554/eLife.31835.015

• Supplementary file 3. Table of Pearson's correlation between environmental variables across sampling sites.
DOI: https://doi.org/10.7554/eLife.31835.016

• Supplementary file 4. Table of Percent of variation explained and p-values for non-interaction linear models for chamber experiment.
DOI: https://doi.org/10.7554/eLife.31835.017

• Supplementary file 5. Table of P-values of interaction term in linear model for chamber experiment.
DOI: https://doi.org/10.7554/eLife.31835.018

• Supplementary file 6. Table of Percent of variation explained and p-values for non-interaction linear models for drought experiment.
DOI: https://doi.org/10.7554/eLife.31835.019

• Supplementary file 7. Table of P-values of interaction term in linear model for drought experiment.
DOI: https://doi.org/10.7554/eLife.31835.020

• Supplementary file 8. Table of $Q_{st}$-$F_{st}$ results.

DOI: https://doi.org/10.7554/eLife.31835.021

• Transparent reporting form
DOI: https://doi.org/10.7554/eLife.31835.022

## Major datasets

The following dataset was generated:

| Author(s) | Year | Dataset title | Dataset URL | Database, license, and accessibility information |
|---|---|---|---|---|
| Supple MA, Bragg JG, Broadhurst LM, Nicotra AB, Byrne M, Andrew RL, Widdup A, Aitken NC, Borevitz JO | 2017 | Eucalyptus melliodora Genotyping-By-Sequencing (GBS) | https://www.ncbi.nlm.nih.gov/bioproject/PRJNA413429/ | Publicly available at NCBI BioProject (Accession no. PRJNA413429) |

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
