## [Decision Letter]

Thank you for submitting your article "Landscape genomic prediction for restoration of a Eucalyptus foundation species under climate change" for consideration by *eLife*. Your article has been favorably evaluated by Ian Baldwin (Senior Editor) and three reviewers, one of whom is a member of our Board of Reviewing Editors. The following individuals involved in review of your submission have agreed to reveal their identity: Angela Hancock (Reviewer #2).

The reviewers have discussed the reviews with one another and the Reviewing Editor has drafted this decision to help you prepare a revised submission.

Summary:

This manuscript shows an impressive combination of genomic, ecological and common garden work within a perennial non-model system. All the reviewers and editors agreed that this work will help to set the standard for how genomics and conservation ecology should be blended.

Essential revisions:

However, given the belief that this work will help to set the standard, there are some key requests from the reviews below that need to be addressed. This will help to elevate the potential impact of this manuscript going forward. These requests are as follows.

1) Break down F_st_ by sub-genomic/gene chunks to assess variation around the genomic mean and how this is influenced by environmental drivers. If this is not possible explain why.

2) Compare and contrast environmental drivers/variables in this system versus other published system to place the study in the proper broad context.

3) Rework the figures and writing to a higher quality of presentation.

Reviewer #1:

This is a potentially interesting manuscript linking genomic variation to functional variation to potential conservation planning. However the manuscript is not aided by the writing and figures. The figures seem very crude and often don't convey the message cleanly. Then this is backed up by incomplete writing that does not have transitions or explanations leaving numerous thoughts and topics to seem to randomly appear and disappear without a coordinated presentation. There is a coordinated thought in this work but that thought needs to be transmitted through precise writing that walks a generalist audience through each and every line of thought.

Reviewer #2:

The authors produced a GBS dataset in the non-model foundation species *Eucalyptus melliodora* using data from 275 trees from 37 sampling sites. They used these data to infer the rate of isolation by distance and to model the effects of environmental variables and geographic distance on F_st_ among populations and compared this model to a model using projected climatic data. They also assessed phenotypic variation in seedlings grown in simulated 'current' and 'future' conditions and found variation in responses among sites.

The main message of the paper is that the genotypes that are ecologically the best-fit to a particular location can come from a much broader geographic range than expected. This has implications that are highly relevant to the design of restoration projects in this species (and perhaps more generally). It suggests that optimal restoration should include a relatively diverse set of individuals.

Given the difficulty implicit in working with non-model systems, I found this to be an impressive study, which will likely shape future research in restoration ecology in diverse species.

Reviewer #3:

The paper surveys genomic variation, using approx. 9400 SNPs in 275 individual trees from 37 sites across the range of *Eucalyptus melliodora*. Key environmental variables are combined with these genomic data to determine resilient populations in the face of climate change. The paper is well written, and the findings are clearly presented. The Introduction section introduces the issue well and puts the current study in the context based on the background of previous studies. The methodology is presented precisely and exhaustively. The results are well presented and discussed. I have only a few general questions, some of which could be beyond the scope of the paper. The questions could be ether addressed in Methodology and Results, or only mentioned (explained) in the Discussion section. The work is certainly of value to readers of the journal.

With respect to the identified environmental drivers/variables, is there any evidence in the literature that similar studies (if any) of related or other species have same drivers? It appears the "usual culprits" are common across most species/plants: some aspect of rainfall, temperature, soil aspect (subsection “Genomic Analyses”, third and fourth paragraphs), although the below-ground variables are difficult to measure?

Subsection “Genomic Analyses”, sixth paragraph – some logic behind the biggest differences in regions may include the following: The northern end is situated at a relatively high altitude (1,000 to 1,300 m) in a summer-rainfall zone and largely on basaltic soils. Altitude is associated with a whole suite of bioclimatic variables important for tree growth. The central region is the most complex in terms of geology and rainfall. Rainfall is also variable, generally decreasing with decreasing altitude and towards the west of the distribution. The southern inland section is in a winter-rainfall zone, has a wide altitudinal range (400 to 1,000 m) with the higher elevation areas receiving occasional falls of snow in winter, although the snow seldom persists on the ground for more than a few days. The underlying geology includes granodiorite and dolerite in the higher-altitude areas, siltstone, shale, and quartzite in the lower areas, plus small areas of basalt.

Subsection “Genomic Analyses” – statistical modelling and inference problems with sample sizes substantially smaller than the number of available covariates are challenging – often referred to as large p small n problem. Furthermore, the problem is more complicated when we have multiple correlated responses. Out of interest, how do your models deal with such an issue i.e., large number of markers on a small number of phenotype/environmental variables?

Discussion, third paragraph – can the authors comment on the role of co-adapted gene complexes?

With respect to F_st_ estimates – perhaps the authors can think more about this but may not be related specifically to this study. For example, it is now common for population geneticists to estimate F_st_ for a large number of loci across the genome. However, one surprising result of such F_st_ scans is the often high proportion (>1% and sometimes >10%) of outliers detected, and this is often interpreted as evidence for pervasive local adaptation. There have been arguments that suggest that correlated co-ancestry inflates the neutral variance in F_st_ when compared to its expectation under an island model of population structure. As a consequence, I find that results of F_st_ scans have become somewhat difficult to interpret in terms of their genome distribution and their effects on phenotype. Perhaps the authors can comment on this in light of their F_st_ results.

---

## [Author Response]

Essential revisions:However, given the belief that this work will help to set the standard, there are some key requests from the reviews below that need to be addressed. This will help to elevate the potential impact of this manuscript going forward. These requests are as follows.1) Break down F_st_ by sub-genomic/gene chunks to assess variation around the genomic mean and how this is influenced by environmental drivers. If this is not possible explain why.

We designed the study to focus on how environment and geography affect the genome as a whole, rather than the impact on specific genes. Given the low genomic resolution of the GBS data in a species where linkage disequilibrium extends only a few hundred base pairs, we feel that a gene level analysis would not provide additional insights. To address this important point, we have added text to the Introduction and genomic results that clarifies our intention to focus on the effects on the genome as a whole, rather than on specific genes. We have also added a paragraph to the Discussion regarding a potential follow-up study using whole genome sequencing to identify adaptive alleles.

2) Compare and contrast environmental drivers/variables in this system versus other published system to place the study in the proper broad context.

We have added a discussion comparing the drivers we identified with those identified in niche modelling of *E. melliodora* and within species genetic variation in other systems. Given the different methods and environmental variables used, direct comparisons between studies are difficult, as are insights into broad trends. Some analyses focus on putatively neutral loci, while other focus on putatively adaptive outlier loci. The studies usually include some measures of temperature and precipitation, but the exact variants on these factors are often different between studies. There are numerous non-climate variables available, such as soil or topography, which vary substantially between studies. The different methods and suites of environmental variables used in each study obscure broad trends that may exist.

3) Rework the figures and writing to a higher quality of presentation.

We have revised the text to make it clearer and more readable. We have redesigned a number of the figures for a cleaner presentation.

Reviewer #3:The paper surveys genomic variation, using approx. 9400 SNPs in 275 individual trees from 37 sites across the range of Eucalyptus melliodora. Key environmental variables are combined with these genomic data to determine resilient populations in the face of climate change. The paper is well written, and the findings are clearly presented. The Introduction section introduces the issue well and puts the current study in the context based on the background of previous studies. The methodology is presented precisely and exhaustively. The results are well presented and discussed. I have only a few general questions, some of which could be beyond the scope of the paper. The questions could be ether addressed in Methodology and Results, or only mentioned (explained) in the Discussion section. The work is certainly of value to readers of the journal.With respect to the identified environmental drivers/variables, is there any evidence in the literature that similar studies (if any) of related or other species have same drivers? It appears the "usual culprits" are common across most species/plants: some aspect of rainfall, temperature, soil aspect (subsection “Genomic Analyses”, third and fourth paragraphs), although the below-ground variables are difficult to measure?

This has been added as suggested. See Essential revision #2 above for more details.

Subsection “Genomic Analyses”, sixth paragraph – some logic behind the biggest differences in regions may include the following: The northern end is situated at a relatively high altitude (1,000 to 1,300 m) in a summer-rainfall zone and largely on basaltic soils. Altitude is associated with a whole suite of bioclimatic variables important for tree growth. The central region is the most complex in terms of geology and rainfall. Rainfall is also variable, generally decreasing with decreasing altitude and towards the west of the distribution. The southern inland section is in a winter-rainfall zone, has a wide altitudinal range (400 to 1,000 m) with the higher elevation areas receiving occasional falls of snow in winter, although the snow seldom persists on the ground for more than a few days. The underlying geology includes granodiorite and dolerite in the higher-altitude areas, siltstone, shale, and quartzite in the lower areas, plus small areas of basalt.

This is an interesting point and highlights how complex environmental adaptation is. Using spatial climate and soil data, we are attempting to include this complexity in our model of how genetics associates with environment.

Subsection “Genomic Analyses” – statistical modelling and inference problems with sample sizes substantially smaller than the number of available covariates are challenging – often referred to as large p small n problem. Furthermore, the problem is more complicated when we have multiple correlated responses. Out of interest, how do your models deal with such an issue i.e., large number of markers on a small number of phenotype/environmental variables?

While we genotyped a large number of markers, the modelling was done using summary statistics (i.e. genetic distance between samples and F_st_ between sites). We focused on the effects on the genome as a whole, rather than the impact on specific loci, in part because of the issues mentioned here.

Discussion, third paragraph – can the authors comment on the role of co-adapted gene complexes?

Since we did not analyze individual genes, our study does not shed any light on co-adapted gene complexes.

With respect to F_st_ estimates – perhaps the authors can think more about this but may not be related specifically to this study. For example, it is now common for population geneticists to estimate F_st_ for a large number of loci across the genome. However, one surprising result of such F_st_ scans is the often high proportion (>1% and sometimes >10%) of outliers detected, and this is often interpreted as evidence for pervasive local adaptation. There have been arguments that suggest that correlated co-ancestry inflates the neutral variance in F_st_ when compared to its expectation under an island model of population structure. As a consequence, I find that results of F_st_ scans have become somewhat difficult to interpret in terms of their genome distribution and their effects on phenotype. Perhaps the authors can comment on this in light of their F_st_ results.

We found low genome wide F_st_ values, but would expect that F_st_ scans would find outlier loci, either by random chance or possibly due to loci linked to local adaptation. Our results could be used to select samples to minimize issues, such as co-ancestry, and provide a more powerful dataset for a future study to identify loci underlying potential adaptation.